# A Knowledge Representation System for the Indian Stock Market

Bikram Pratim Bhuyan [1,2,*,†] , Vaishnavi Jaiswal [2,†] and Amar Ramdane Cherif [1,*,†]

1   LISV Laboratory, University of Paris Saclay, 10-12 Avenue of Europe, 78140 Velizy, France
2   School of Computer Science, University of Petroleum and Energy Studies, Dehradun 248007, India
*   Correspondence: bikram23bhuyan@gmail.com (B.P.B.); amar.ramdane-cherif@uvsq.fr (A.R.C.)
†   These authors contributed equally to this work.

**Abstract:** Investors at well-known firms are increasingly becoming interested in stock forecasting as they seek more effective methods to predict market behavior using behavioral finance tools. Accordingly, studies aimed at predicting stock performance are gaining popularity in both academic and business circles. This research aims to develop a knowledge graph-based model for representing stock price movements using fundamental ratios of well-known corporations in India. The paper uses data from 15 ratios taken from the top 50 companies according to market capitalization in India. The data were processed, and different algorithms were used to extract tuples of knowledge from the data. Our technique involves guiding a domain expert through the process of building a knowledge graph. The scripts of the proposed knowledge representation and data could be found here: GitHub. The work can be integrated with a deep learning model for explainable forecasting of stock price.

**Keywords:** artificial intelligence; stock market; knowledge representation and reasoning; fundamental analysis; symbolic AI

## 1. Introduction

The stock market is essential for any developed country, particularly for rapidly expanding economies such as India [1]. The performance of the stock market may be linked not only to the prosperity of the country itself but also to the progress of other emerging economies [2]. A positive correlation exists between a nation's expanding economy and the growth of its stock market. Conversely, if the stock market were to fall, it would negatively impact a country's economic growth. In other words, the stock market's condition has a direct impact on a nation's overall economic growth. However, due to the unpredictability of the stock market, only a small percentage of people actively participate in it. As a result, many people mistakenly believe that trading stocks is similar to gambling. Increasing public awareness about the stock market is crucial to dispel this misconception. Prediction strategies for the stock market can spark the interest of new investors while holding the attention of experienced investors. The positive results that can be achieved using prediction tools can change public perception. Business organizations can benefit from data mining technologies that can forecast future trends and behaviors, providing insights that enable informed and data-driven decision-making. This can be achieved by using data mining and machine learning techniques of artificial intelligence to predict future trends and behaviors, as demonstrated in the literature [3].

Knowledge representation/reasoning is a subfield of symbolic artificial intelligence that aims to automate human-like thinking by developing machines capable of reasoning about the world based on data engines that can understand it [4]. This subfield was founded to automate human-like thinking. Knowledge-based systems can represent real objects, events, relationships, and other aspects of a domain with symbols when provided with a computational model of the area of interest. Describing knowledge is the same process in the mathematical/algorithmic area, regardless of the domain of interest, whether it is the

world as we know it or a completely fictitious system. Knowledge-based systems reason by manipulating the symbols of the computational model, which are stored in the system's knowledge base as domain-specific statements. When the system needs to reason about a particular topic, these statements are accessed. A knowledge base can provide responses to questions about a specific subject, which an application can use to guide its behavior.

A stock represents a portion of a company's ownership stake, and the terms "stock" and "equity" are not interchangeable. The smallest unit of ownership in a company is known as a share in the world of stocks. While it is possible for an individual to own stocks in several different firms, the number of shares they hold in a specific company is fixed (e.g., 10 shares in P, 100 shares in Q, 500 shares in R, etc.). A stock is a more specific term that refers to ownership in a single corporation, while "share" is a more general term. Because stocks can be quickly converted into cash, they are considered liquid assets.

When we use the term "market," we typically mean a place where people meet regularly to buy and sell goods. In the context of buying and selling stocks, the stock market can be defined as a market for both short-term and long-term trading on various exchanges, including the New York Stock Exchange (NYSE) in the United States, the National Stock Exchange (NSE) in India, the Tokyo Stock Exchange (TYO) in Japan, and many more depending on the country.

There are two types of stock markets: public stock exchanges and private equity crowdfunding platforms. According to Wikipedia, "A stock market, equity market, or share market is the gathering of buyers and sellers of stocks, which reflect ownership claims on firms" [5]. After an initial public offering (IPO), investors can purchase or sell shares of a company they are interested in. Companies issue stocks to acquire funds for expanding their operations. Price changes are driven by various factors, including supply and demand.

*1.1. Indian Stock Market*

The history of the stock market can be traced back to the development of stock exchanges, which provide a marketplace for investors to buy and sell equities. For a long time, commodities have been traded in Mumbai, a major commercial port. In the 1850s, a banyan tree in front of Mumbai's Town Hall was used as a meeting place for business transactions [6]. Informal get-togethers were held where cotton was casually traded.

After establishing the Companies Act in 1850, people became interested in corporate security. The Native Share and Stock Broker's Association was established in 1875. In 1875, Bombay's first stock market was established. The Ahmedabad stock exchange, the Calcutta stock exchange, and many others were born due to this development. The Securities and Exchange Board of India (SEBI) was established in 1988 to monitor, regulate, and grow the Indian stock markets. In 1992, it became a fully autonomous and self-governing organization. There are 29 stock exchanges in India, including the National Stock Exchange (NSE), the Over-the-Counter Exchange of India (OICEI), and 21 regional stock exchanges with designated territories. In the wake of SEBI's regulatory tightening, all (except the Calcutta Stock Exchange) were shut down. Corporate councils and CEOs oversee the stock exchange operations.

The Ministry of Finance controls the policies. Most trading occurs on the National Stock Exchange (NSE) and the Bombay Stock Exchange (BSE). Many additional stock exchanges exist in India, but their combined trading volume is so small that they have little impact on the market [7].

1.1.1. Bombay Stock Exchange (BSE)

A Rajasthani Jain businessman named Premchand Roychand founded the publicly traded BSE in 1875. The BSE is the first stock exchange in Asia and the largest in the region. This stock exchange is one of the quickest in the world, transferring information in only six microseconds. As a result, India's business sector has grown. For the last 143 years, it has served as a capital-raising tool and it continues to do so.

It is the mission of the BSE to provide a market where financial instruments, including currencies, stocks, and mutual funds, may be traded in an orderly and transparent manner. The BSE offers several different indexes. BSE SENSEX is one of the most well-known indicators (S&P Bombay Stock Exchange Sensitive Index or simply SENSEX). It consists of 30 well-established and financially secure enterprises. In India, the BSE SENSEX is regarded as the heartbeat of the country's stock market. Internationally, it is traded on EUREX and other BRCS markets (Brazil, Russia, China, and South Africa).

The 'BSE 100', 'BSE 300', 'BSE MIDCAP', 'BSE SMALLCAP', 'BSE Auto', 'BSE Pharma', and 'BSE Metal' are key indexes in addition to the SENSEX [8].

### 1.1.2. National Stock Exchange (NSE)

In 1992, the government instructed a private limited company, NSE, to be formed to increase transparency in the Indian capital market. It is one of India's most important stock exchanges. To "continue to be a leader, build a worldwide footprint, promote the financial well-being of people", the NSE was the country's first dematerialized electronic exchange. When it comes to trading derivatives, the NSE is the finest. In other words, futures and options traders should only trade on the NSE [9].

The most well-known NSE index, the 'NIFTY50', is the source of the data we used in this study. This group includes the top 50 publicly traded corporations in India. Investors often use this index to monitor the Indian capital market. The index comprises 13 major economic sectors in India, namely: financial services, information technology, consumer goods, oil and gas, automobiles, telecommunications, construction, pharmaceuticals, metals, power, cement and cement products, fertilizers and pesticides, and media and entertainment. The respective stocks for each sector are shown in Figure 1.

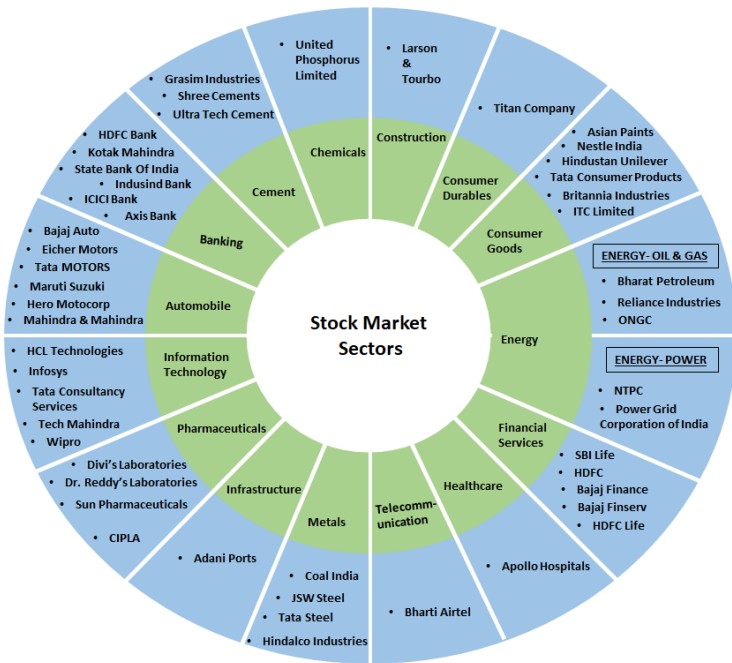

**Figure 1.** Sectors with respective stocks in NIFTY50.

One reason for using this dataset is that the NSE is the finest derivative stock market. While the Bombay Stock Exchange (BSE) allows even substandard firms to list their shares, the National Stock Exchange (NSE) maintains tight listing standards. Over the last decade, the stock market has become more popular, but many individuals are either ignorant of its advantages or unwilling to participate because of the risks involved. Stock market prediction algorithms play an important role in creating apps or websites that regular consumers can visit to gain insight into which firm they should invest in.

In this paper, we first discuss the various techniques involved in predicting the stock market. The following section focuses on the basic terminologies used in this article, as well as the topic of knowledge representation techniques, known as formal concept analysis. We then describe the methodology used in this study. Next, we present the results and their description in a subsequent section. Finally, we conclude this research work with a discussion of the potential future implications of our findings.

## 2. Literature Review

Based on the stock market prediction techniques shown in Figure 2, this literature study shows the various techniques used in stock market prediction.

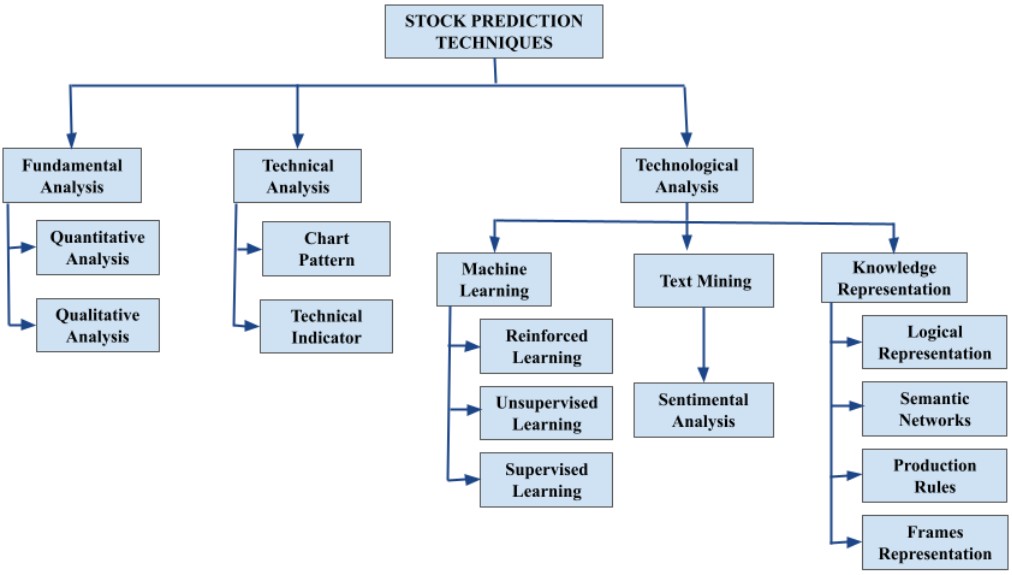

**Figure 2.** Different techniques used in stock prediction.

### 2.1. Fundamental Approach

Numerous fundamental approaches have been tried and tested for stock market prediction. These approaches are discussed here.

The authors of [10] examined a wide range of literature on forecasting the stock market using fundamental, technical, or mixed methods. The analysis involved 122 studies published in academic journals between 2007 and 2018. The study found that understanding stock price movement over the long term may be better achieved through the use of fundamental analysis. According to the research, the most commonly used machine learning algorithms for making market forecasts are support vector machines (SVMs) and artificial neural networks (ANNs). The research also suggests that considering both internal and external factors can lead to more accurate and precise estimates.

Sharma et al. (2017) [11] analyzed a popular regression technique for predicting stock prices based on historical data. They conducted a study on polynomials, KBF, sigmoid, and linear regression. The primary aim of their research was to facilitate brokers and investors to invest more in the stock market. Data mining may identify both revealed and hidden patterns that boost accuracy when conventional and statistical approaches are weak.

Gurvinder et al. [12] (2019) evaluated different machine learning approaches in five important industries, i.e., agriculture, finance, healthcare, education, and engineering. These industries still require greater attention. Nature-inspired computing (NIC) strategies require greater consideration for multiverse optimization techniques. The latest and emerging NIC approaches should also be employed in these areas, and their performance should be analyzed.

Selvamuthu et al. [13] (2019) conducted a study using neural networks based on three different learning algorithms (Levenberg–Marquardt, scaled conjugate gradient, and Bayesian regularization) for predicting the stock market based on tick data and 15-minute data of an Indian company. The study compared the results of these algorithms. For tick data, all three algorithms achieved an accuracy of 99.9%; however, for 15-minute data, the accuracies declined to 96.2%, 97.0%, and 98.9%, respectively. The study found that the average performance of the ANN model was much better than the SVM model. While the LSTM and other neural networks used in this research were effective, recurrent neural networks (RNNs) may offer even greater accuracy, particularly with long short-term memory. Sentiment analysis can also provide an advantage in predicting stock values, which are influenced by the statements and views of famous people.

Chen et al. (2017) [14] aimed to enhance stock market analysis by considering financial factors, environmental conditions, macroeconomics, and financial news to establish a fundamental approach to stock market forecasting. The technique developed in this research has the potential to improve the quality of investment decisions by increasing the reliability of predicted stock trading signals. However, the research did not consider the meaning of phrases in financial news headlines, making it difficult to oversee the system's learning process. Inspecting the gathered execution contexts requires the supervision of an experienced software engineer.

Pathak et al. (2018) [15] used a hybrid model, created by merging various methods. This research combined a quantitative method with a qualitative one. The quantitative analysis relied on past data, while qualitative analysis utilized new sources, such as news articles and Twitter feeds to gauge public opinion. The study covered machine learning, sentiment analysis, and fuzzy logic. If the News Sentiment is positive or the Stock Prediction value is excellent, THEN the Stock Faith will be high, and vice versa. The suggested approach can potentially create a real-time trading model that calculates total returns or investments.

### 2.2. Technical Approach

2.2.1. Chart Pattern

Myoung et al. (2016) [16] conducted a finance research study based on chart pattern analysis. This strategy has not received much attention despite the insights that can be gleaned from charts depicting price changes. The researchers proposed an algorithm for the creation of rule-based chart patterns. They defined a rule-based chart pattern search as an optimization problem and developed a genetic algorithm. The study found that the best-performing identified patterns on the Korean stock market are rising-support patterns and employ characteristics at a high degree of abstraction.

Wit et al. (2019) [17] created an innovative approach to investment decisions that may assist novice investors using the large volume and high-velocity characteristics of Big Data provided by the New York Stock Exchange. The authors utilized publicly available data and machine learning to automate chart pattern analysis in a way that shows weekly trading is a more practical method for nonprofessional investors, as opposed to using high-frequency trading tactics.

Zhu (2018) [18] proposed a new domain-specific language, called FCPL, for expressing technical patterns using a set of five guidelines. The grammar of the proposed language is formally characterized using EBNF. The primary objective of FCPL is to become a simple, comprehensive, expressive, and reusable domain-specific language for creating chart patterns in financial trading. A programming language can be derived from the definitions of the chart patterns provided by FCPL. The implementation details of a chart pattern are abstracted away from the pattern's specification in FCPL. The results indicate that the suggested approach can provide a higher index return with fewer trades. In terms of compounded returns, it even outperforms the best mutual funds.

Y. -P. Wu et al. (2012) [19] combined the sequential chart pattern, the K-means, and the AprioriAll algorithm to anticipate stock movement. A sliding window is used to condense

the stock price series into charts. The K-Means algorithm then clusters the charts into recognizable patterns. AprioriAll may then be used to extract common customs from the chart sequences.

Siriporn et al. (2019) [20] utilized the chi-square automatic interaction detector (CHAID) algorithm to analyze the relationships between characteristics and develop a trading strategy. The objective was to establish trading techniques that generate significant returns on investment. Ten equities traded on the Stock Exchange of Thailand were used to evaluate the method against eight widely used trading techniques. The results demonstrate the viability of the proposed method in real-world securities trading and its ability to generate trading strategies that outperform other strategies.

### 2.2.2. Technical Indicator

Ahmar et al. (2017) [21] developed a stock market technical indicator evolution using the Sutte indicator. The article discusses the Sutte indicators used in stock trading to assist investors in deciding whether to purchase or sell shares. The effectiveness of the Sutte indicator was evaluated in comparison to two other forms of technical analysis: the simple moving average (SMA) and the moving average convergence/divergence (MACD). The accuracy with which stock data could be predicted was compared using the mean squared error (MSE), mean absolute deviation (MAD), and mean absolute percentage error (MAPE). The findings indicate that the Sutte indicator has the potential to be a tool for predicting changes in stock prices. According to the MSE, MAD, and MAPE, the Sutte indicator is more reliable than the SMA and MACD indicator methods.

Alfonso et al. (2020) [22] proposed using non-linear technical indicators as inputs to non-linear models via combinatorial techniques. This methodology resulted in better outcomes than non-linear forecasting algorithms, such as neural networks that employ all of the relevant variables. The suggested approach provides a practical alternative for estimating predictions for all potential combinations of many technical indicators (independent variables).

### 2.3. Machine Learning

### 2.3.1. Reinforced Learning

Meng et al. (2019) [23] demonstrated that reinforcement learning can significantly outperform baseline models in predicting accuracy and trading profitability when implemented in the appropriate setting. However, some studies have shown that reinforcement learning may not perform well when there are significant differences in pricing patterns between the training and testing data. Additionally, there have been few studies that compared the effectiveness of reinforcement learning models to other models, such as autoregressive integrated moving average (ARIMA), deep neural networks, recurrent neural networks, or state space models, making it challenging to draw direct comparisons between them.

The method proposed by Ezzeddine et al. (2021) [24] combines deep learning with deep reinforcement learning in a three-layer architecture to enable efficient stock trading using ensembles. The proposed approach utilizes multiple ensemble steps to provide its intra-day trading strategy. First, historical market data are encoded into GAF images, and hundreds of deep learning decisions from multiple CNNs trained with this data are stacked into the images. These images are then used as input for a reinforcement meta-learner classifier.

### 2.3.2. Unsupervised Learning

In [25], collaborative filtering was used to forecast the daily transaction volume in the Nifty 50 equities index. This study utilizes data from the NSE stock market to develop a recommender system using collaborative filtering. The daily trading activities of 50 firms across ten industries are considered, with the top 5 companies in each industry taken into account. The results demonstrate that increasing the user base of the model can enable

the processing of large datasets. Additionally, the study shows that the k-NN algorithm performs better than other alternatives.

Song et al. (2020) [26] demonstrated how to employ supervised learning algorithms to spot suspicious trades associated with stock market manipulation. Song and colleagues used an example of stock manipulation in 2003 and employed CART, CI-trees, C5.0, RF, NB, NN, SVM, and kNN to classify skewed data. The empirical data demonstrated that Naive Bayes had the highest performance of all the learning techniques, with an F2 of 53% (sensitivity and specificity of 89% and 83%, respectively). This research explored using supervised learning algorithms to identify stock market manipulation; the authors provided a case study and utilized the algorithms to develop models for predicting transactions that are possibly related to market manipulation.

### 2.3.3. Supervised Learning

K. Golmohammadi et al. (2014) [27] used supervised learning algorithms to detect potentially fraudulent stock market trades. To do this, the authors examined an example of stock manipulation in 2003. To classify skewed data, they used CART, CI-trees, C5.0, RF, NB, NN, SVM, and kNN. The empirical data show that Naive Bayes has the highest performance of all the learning techniques, with an F2 of 53% (and sensitivity and specificity of 89% and 83%, respectively). This study explored using supervised learning algorithms to identify stock market manipulation. To demonstrate the efficacy of these algorithms, the authors offered a case study and utilized it to develop a model for anticipating trades that may indicate market manipulation.

Khattak et al. (2019) [28] proposed a stock trend prediction system by employing a supervised machine learning approach, namely the K-nearest neighbor classifier. Their emphasis was on reducing data sparseness in the collected datasets by conducting outlier identification on the acquired dataset for dimensionality reduction. The experimental findings support the efficiency of the suggested method. However, they concluded that additional machine learning classifier experiments are needed and that future work should focus on using deep learning methods, such as LSTM, RNN, and CNN on various datasets.

### 2.4. Text Mining

Sentimental Analysis

Kesavan et al. (2020) [29] combined polarity ratings collected from sentiment research into stock price forecasting models with historical stock time-series data. Since the events and psychology of the investors directly impact the stock market, the suggested technique delivers accurate results.

Alzazah et al. (2020) [30] discussed the efficiency of text mining and sentiment analysis methodologies in forecasting market movements. By comparing several ML approaches, such as SVM and decision tree, and deep learning models, such as LSTM and CNN, they explored some of these models' limits and future work and disputed the best outcome achieved by each one of these models. Ultimately, the proposed survey showed the need to enhance prediction methods by doing certain things, such as adding the structural information, taking into account event sentiments analysis, using more effective expanded lexicons, amping up the number of collected news, lengthening the training period, applying the deep learning models, adding different sources of information, upgrading the sentiment analysis task by amping up the words that may affect stock movements more, and employing unsupervised learning.

In a study by Rajendiran et al. (2021) [31], the authors examined the assessment of various traditional stock market prediction models using dynamic analysis. The developed stock market categorization model did not improve accuracy, as shown by the findings of the survival analysis. The dynamic analysis approach did not lead to more accurate predictions, and the Q-learning model did not reduce prediction time. Although conventional methods have proved to be reliable, several machine learning and classification strategies were used to improve the efficiency and accuracy of stock market predictions.

### 2.5. Formal Concept Analysis (FCA)

Škopljanac-Mačina et al. (2014) [32] proposed a method to facilitate the use of formal concept analysis in classrooms and computerized learning systems. The purpose of these tools is to assist educators in ensuring that test questions cover the required content by reviewing question pools. Formal concept analysis takes a predetermined set of questions as input. The resulting concept lattice provides teachers with a visual representation of the optimal exam design and other useful information. Topological sorting algorithms can be applied to the concept lattice to determine which concepts are more challenging for students to understand. Lastly, the authors discuss how concept lattices can serve as a foundation for ontology development.

Shah et al. (2019) [33] provided a comprehensive literature review of state-of-the-art algorithms and methodologies commonly used for stock market prediction, highlighting current challenges that require greater attention and suggesting options for future improvement and research. Machine learning, expert systems, and Big Data are among the technologies utilized in the development of today's stock markets, and they work together to assist traders and investors in making better decisions.

Instructions for conducting formal concept analysis were provided by Ignatov et al. (2015) [34]. This branch of mathematics permits the formalization of ideas as object-attribute units of human cognition and analysis. The goal of this introductory text is to provide the reader with the theoretical foundations and a variety of practical examples and exercises needed to apply FCA.

### 2.6. Knowledge Graphs

Financial and stock market analyses are just two examples of the many fields where knowledge graphs are shown to be invaluable tools. By representing data in a structured and semantically meaningful fashion, knowledge graphs can help analysts perform more precise and efficient analyses.

The authors of [35] presented a knowledge graph-based method for predicting the movements of well-known companies' stock prices. Companies' financial data, news stories, and other pertinent information were compiled into a knowledge graph, and then machine learning algorithms were used to draw conclusions and predict stock market movements. The authors further assessed the methodology by contrasting their forecasts with historical stock market information. Evidence from this study demonstrates that the knowledge graph approach to stock market prediction is superior to more conventional techniques and may yield useful information for traders and analysts.

Predicting stock price trends in China's stock exchange market was investigated in [36]. The authors proposed a unified framework that incorporates a knowledge graph and represents the interconnections between various financial entities with deep learning models, such as convolutional neural networks and long short-term memory networks. The proposed approach was compared to the accuracy of predictions made by conventional machine learning models; the results show that the proposed framework performs better. Stock price trend forecasting using deep learning and knowledge graphs is demonstrated to be feasible, and the paper suggests that this methodology may be extended to other financial markets.

Enterprise knowledge graphs and news sentiment analysis were used to predict stock prices in [37]. The authors suggested combining these two methods to better anticipate stock values. The corporate knowledge graph links products, workers, and customers and is created using financial reports, news announcements, and social media. Entities and relationships are extracted from text using natural language processing to create a graph-based business representation. News sentiment analysis employs natural language processing to evaluate enterprise-related news stories, with each article's sentiment score determined by a lexicon-based approach. The authors tested their methodology using Chinese stock market data and estimated five company stock values over three months using the method. The proposed method outperformed support vector regression and

random forest regression in terms of prediction accuracy. Other trials examined how various variables affected prediction performance, with the corporate knowledge graph and news sentiment analysis having the most impact on forecast accuracy. The prediction performance also depends on the knowledge graph size and news article duration. The article concludes that business knowledge graphs and news sentiment analysis can be used to forecast stock prices and advises applying the method to other financial markets, using social media data and analyst reports.

The authors of [38] built a knowledge graph to show the relationships between stocks, industries, and financial reports. This knowledge graph helped them find connected stocks and mutation points, which are important price changes. The authors then estimated stock prices using a convolutional neural network (CNN). The CNN predicted stock prices based on daily stock prices and financial indicators. To boost prediction accuracy, the authors added knowledge graph-identified stocks and mutation sites to the CNN model. The authors tested their methodology using Chinese stock market data and compared their system to baseline methods, such as machine learning and CNN models. The framework outperformed the baseline models in terms of prediction accuracy. The authors also tested how different factors affected prediction performance. The knowledge graph's related stocks and mutation points significantly improved prediction accuracy. They also found that the CNN model's prediction performance depended on the historical data length. The article concluded that knowledge graphs and deep learning can forecast stock prices. To improve prediction accuracy, the authors suggested adding news articles and social media data.

The authors of [39] intend to create a model that can accurately predict the impact of electronic word-of-mouth (eWoM) on Reddit posts and identify the factors that contribute to their success. The article emphasizes the importance of eWoM in marketing and the growing role of social media platforms, such as Reddit, for knowledge sharing. The authors presented their machine learning model, which predicts Reddit post performances by analyzing post content, engagement data, and user behavior. They tested their model on 25,000 Reddit posts and compared the predicted post-success to the platform performance, achieving an accuracy of 84%. The authors also examined the success elements of Reddit and found that emotional language, contentious issues, and involvement with prominent subreddits performed better. They also discovered that publishing at specific times and days increased engagement. The report concluded by discussing the potential marketing and social media applications of their methodology for optimizing social media marketing efforts and Reddit promotional content targeting, as well as its potential use in other social media networks.

The authors of [40] presented a new paradigm for analyzing social media sentiment that considers both time and location. A novel sentiment analysis approach that accounts for geographical and temporal features of social media data shows how sentiment fluctuates throughout time and geography. The paper emphasized the importance of sentiment analysis in social media and the drawbacks of existing methods that ignore spatial and temporal data. The authors then introduced their method, which employs machine learning and geographical and temporal analyses to identify sentiment trends in geography and time. The authors evaluated their method using around 500,000 American tweets. It detected subtleties better than prior sentiment analysis methods. Their technique examined COVID-19 tweet sentiments over time and geography. They found variations in pandemic mood throughout the nation, which changed with events and progress. Marketing, public opinion analysis, and crisis management concluded the research. The authors recommended utilizing their technique to find sentiment trends and patterns in social media data to aid decision-making in these industries.

This collection of papers demonstrates the potential of knowledge graphs for stock market analysis and highlights the possibility for further research in this area. Nonetheless, stock market analysis is a complicated and varied discipline; therefore, more investigation into the strengths and weaknesses of knowledge graphs in this area is warranted.

It is important to note that various machine learning and deep learning algorithms for categorization were developed based on knowledge graphs provided by domain experts. Our method leads the domain expert through the graph construction process by utilizing algorithms that extract tuples of knowledge from the data.

## 3. Terminologies Used

Table 1 is the symbol table used to refer to various notations and symbols in this paper.

**Table 1.** Notations used in the article.

| Interpretation | Hieroglyph |
| :---: | :---: |
| Current Market Price | $\kappa_{MP}$ |
| Market Capital | $\kappa_{MC}$ |
| Return on Equity Ratio | $\psi_{ROE}$ |
| Price to Earning Ratio | $\psi_{PE}$ |
| Price to Book Value Ratio | $\psi_{PB}$ |
| Net Income | $\kappa_{NE}$ |
| Shareholder's Equity | $\kappa_{SE}$ |
| Price per Share | $\kappa_{PS}$ |
| Total Number of shares | $\kappa_{TS}$ |
| Share Value or Market value | $\kappa_{MV}$ |
| Book Value | $\psi_{BV}$ |
| Number of Shares | $\kappa_{No.s}$ |
| Tangible Assets | $\kappa_{TA}$ |
| Liabilities | $\kappa_L$ |
| Price of Fixed Assets | $\kappa_{PFA}$ |
| Price of Current Assets | $\kappa_{PCA}$ |
| Earning per Share | $\psi_{EPS}$ |
| Net Profit | $\psi_{NP}$ |
| Preferred Dividend | $\kappa_{PD}$ |
| Number of Common Shares | $\kappa_{No.cs}$ |
| Debt to Equity Ratio | $\psi_{DE}$ |
| Price to Sales Ratio | $\psi_{PS}$ |
| Sales per Share | $\kappa_{SS}$ |
| Current Ratio | $\psi_{CR}$ |
| Quick Ratio | $\psi_{QR}$ |
| Current Assets | $\kappa_{CA}$ |
| Liquid Assets | $\kappa_{LA}$ |
| Dividend Yield | $\psi_{DY}$ |

**Table 1.** *Cont.*

| Interpretation | Hieroglyph |
|---|---|
| Dividend per Share | $\kappa_{DS}$ |
| Price/Earning to Growth Ratio | $\psi_{PEG}$ |
| Growth Rate | $\kappa_{GR}$ |
| Return on Asset Ratio | $\psi_{ROA}$ |
| Total Assets | $\kappa_{TA}$ |
| Interest Coverage Ratio | $\psi_{ICR}$ |
| Earning Before Interest, Taxes, Depreciation, and Amortization | $\kappa_{EBIDTA}$ |
| Asset Turnover Ratio | $\psi_{ATR}$ |
| Net Sales | $\kappa_{NS}$ |

*Market Parameters*

The different parameters used for the study as discussed below:

1.  $\kappa_{MP}$—Current market value of a stock traded on various exchanges, such as NSE, BSE, and others. It is the price one frequently sees mentioned in blogs, on the news, investment platforms, media networks, and finance domains.
2.  $\kappa_{MC}$—The market capital refers to the overall value of all of the shares of stocks for that company. It is calculated by multiplying the current market price of each share by the number of outstanding company shares. It is often used to calculate a company's size and evaluate the financial performance of other various-sized companies.

$$\kappa_{MC} = \kappa_{PS} \times \kappa_{TS} \tag{1}$$

3.  $\psi_{ROE}$—It is the return rate for common stockholders who possess shares in a company's shareholdings. It measures a company's profits and how effectively it makes those profits.

$$\psi_{ROE} = \frac{\kappa_{NI}}{\kappa_{SE}} \tag{2}$$

A very high value of ROE, which is caused by a high net income compared to equity, indicates that a company's performance is good; however, if a high $\psi_{ROE}$ is due to high equity, it means that the company is at risk.

4.  $\psi_{PE}$—This ratio is the most widely used ratio to determine whether a share is overvalued or undervalued. To calculate the $\psi_{PE}$ ratio, we divide the current market price per share by the company's earnings per share (EPS). A lower $\psi_{PE}$ means the stock is undervalued, which is good for buying the stock at that point in time.

$$\psi_{PE} = \frac{\kappa_{PS}}{\psi_{EPS}} \tag{3}$$

5.  $\psi_{BV}$—If we sell all assets and pay all liabilities, then the remaining money is the $\psi_{BV}$ of a company.

$$\psi_{BV} = \frac{\kappa_{TA} - \kappa_{L}}{\kappa_{PFA} + \kappa_{PCA}} \tag{4}$$

$$\psi_{BV}/share = \frac{\psi_{BV}}{\kappa_{No.S}} \tag{5}$$

6.  $\psi_{PB}$—$\psi_{PB}$ is the ratio of the share price divided by the $\psi_{BV}/share$. A lower $\psi_{PB}$ can mean that either a company is underestimated, something is wrong within the

company, or the investor is paying extra for what would remain if the company suddenly went bankrupt.

$$\psi_{PB} = \frac{\kappa_{MV}}{\psi_{BV}/share} \quad (6)$$

7. $\psi_{EPS}$—It is the profit amount allocated to individual outstanding shares of a company's common stock.

$$\psi_{PB} = \frac{\psi_{NP} - \kappa_{PD}}{\kappa_{No.CS}} \quad (7)$$

Preferred dividend—a dividend that is issued to, or paid on, a company's preferred shares.

8. $\psi_{DE}$—It is the total debt divided by the equity of a non-current. Ideally, $\psi_{DE} < 1$ is a comfortable position. In tougher times, companies with low $\psi_{DE}$ (retail, software) survive better.

$$\psi_{DE} = \frac{Total.\kappa_L}{Total.\kappa_{SE}} \quad (8)$$

The $\psi_{DE}$ ratio can be used to see the amount of leverage that a company uses.

9. $\psi_{PS}$—A $\psi_{PS}$ ratio compares a company's stock price to its revenue generated. This ratio shows how much investors are willing to invest per rupee sales for a stock.

$$\psi_{PS} = \frac{Market.\kappa_{PS}}{Annual.\kappa_{SS}} \quad (9)$$

The $\psi_{PS}$ ratio is at its greatest value when comparing companies within the same sector.

10. $\psi_{CR}$—$\psi_{CR}$ is a liquidity ratio that indicates a company's financial position; it measures the company's ability to meet its financial obligations that are due within a year (short-term).

$$\psi_{CR} = \frac{\kappa_{CA}}{Current.\kappa_L} \quad (10)$$

If the $\psi_{CR}$ is less than 1, it is not a good sign, whereas a current ratio of 3.0 signifies that the company can pay its liabilities 3 times faster.

11. $\psi_{QR}$—It is an indicator of short-term liquidity and the company's ability to meet its financial obligations by majorly using liquid assets (considers highly-liquid assets or cash equivalents as part of the current assets).

$$\psi_{QR} = \frac{\kappa_{LA}}{Current.\kappa_L} \quad (11)$$

The higher the $\psi_{QR}$, the better the company's financial health and liquidity.

12. $\psi_{DY}$—The amount of dividend that a company pays each year divided by the price per share; it is expressed in percentage.

$$\psi_{DY} = \frac{Annual.\kappa_{DS}}{\kappa_{PS}} \quad (12)$$

13. $\psi_{PEG}$—It is a stock's P/E ratio divided by its earnings growth rate for a specific time period, i.e., the relationship between growth and the P/E ratio. It is an indicator of a stock's true value; similar to P/E, the lesser value represents a stock that may be undervalued given its future earnings expectations. In a fairly valued company, the P/E ratio and expected growth will be almost equal, making the PEG ratio = 1.0; a PEG below 1.0 means that the company is undervalued, and 1.0 means that the company is overvalued (for example, if company A has P/E = 22 and a growth rate of 20%).

$$\psi_{DY} = \frac{Annual.\kappa_{DS}}{\kappa_{PS}} \quad (13)$$

For example if company A has P/E = 22 and a growth rate of 20%, then PEG = 22/20 = 1.1. Company B, has P/E = 30 and a growth rate of 50%; thus, PEG = 30/50 = 0.6. Even though company A has less P/E, its growth rate is less than company B.

14.  $\psi_{ROA}$—It indicates how much profit a company is generating in relation to its assets. It shows how well a company is performing.

$$\psi_{ROA} = \frac{\kappa_{NE}}{\kappa_{TA}} \tag{14}$$

The greater the return value, the more productive and efficient the company is.

15.  $\psi_{ICR}$—A company's ability to honor its debt payment.

$$\psi_{ICR} = \frac{\kappa_{EBIDTA}}{Expense} \tag{15}$$

EBITDA—earnings before interest, taxes, depreciation, and amortization. The greater the ratio, the better the rest; it may vary from industry to industry.

16.  $\psi_{ATR}$—It measures how efficiently a company uses its assets to generate revenue. It is calculated as revenue divided by average assets.

$$\psi_{ATR} = \frac{\kappa_{NS}}{Average.\kappa_{TA}} \tag{16}$$

*3.1. Knowledge Representation*

Homo Sapiens, or humans in general, are the most evolved species in the animal kingdom. Humans have come a long way from building stone weapons to building supercomputers. Humans excel in understanding, reasoning, and interpreting knowledge. They possess knowledge and act accordingly. Knowledge representation (KR) and reasoning are key components for machines to perform these tasks. Knowledge representation is an artificial intelligence field that focuses on representing knowledge present in the world, in a format that computers can use to solve complex tasks, drawing from psychology to understand how humans solve problems and represent knowledge in a way that facilitates the design and construction of complex systems.

There are four techniques of knowledge representation.

1. Logical representation.
2. Semantic networks.
3. Production rules.
4. Frame representation.

3.1.1. Logical Representation

Logical representation involves drawing conclusions based on conditions and representing facts to machines. It uses proper syntax and rules to present knowledge to machines. There should be no ambiguity in the syntax; it must deal with prepositions.

Logical representation is the foundation of programming and enables logical reasoning. There are two types of logical representation: propositional logic and first-order logic. Propositional logic, also known as statement calculus, operates on the Boolean method (true or false). First-order logic, also called first-order predicate calculus logic, is a more advanced version of propositional logic; it represents objects in quantifiers and predicates.

Logical representation is the most basic form of representation, but its strict representation makes it tough to work with and less efficient at times.

3.1.2. Semantic Network Representation

Semantic networks represent knowledge using graphs where objects are represented as nodes, and the relationships between the objects are represented through arcs. They show how objects are connected. Relationships in semantic networks can be of two types, i.e., IS-A (inheritance) or KIND-OF relations. These relations are more natural than logical

but are not intelligent and, thus, depend on the system's creator. These are simple and easy to understand.

### 3.1.3. Production Rules

Production rules are the most commonly used method for representing knowledge in artificial intelligence. They consist of condition–action pairs, similar to an if–else system; if a condition exists, the production rules fire and action is taken. The condition determines which rule or rules may apply. Production rules are comprised of three parts: the production rule itself, working memory, and the recognized art cycle. The production rule evaluates the condition and applies the corresponding rules. Working memory stores information about the current problem-solving process and rules for writing knowledge to the working memory. The action part follows the recognized art cycle step in problem-solving. Production rules are easily modified and expressed in natural language. However, they may be inefficient as many rules are active during a single program execution and do not store past data, thus not gaining any experience.

### 3.1.4. Frame Representation

Frame representation, as the name suggests, is a record-like structure that stores attributes and their assigned values using 'slot' and 'filler'. The filler is the value of the slot and can be of any data type or shape. Frames can be divided into structures and substructures, enabling similar data to be combined into groups. Similar to other data structures, frames can be visualized and manipulated easily. Frames were derived from semantic networks and have evolved into objects and classes.

### *3.2. Formal Concept Analysis*

Mathematical Definitions

**Definition 1.** *A formal context* $(G, M, I)$ *is defined as a triple where*

- *$G$ represents a set of objects.*
- *$M$ represents a set of attributes.*
- *$I$ represents a binary relation.*

$$I \subseteq G \times M \tag{17}$$

**Definition 2.** *Derivation Operators for $A \subseteq G$ and $B \subseteq M$ where $A$ is a set of objects and $B$ is a set of attributes.*

- *$A' = \{ m \in M \mid \forall g \in A : g \mid m \}$.*
- *$B' = \{ g \in G \mid \forall m \in B : g \mid m \}$.*

**Definition 3.** *Proposition for $A, C \subseteq G$*

- *$A \subseteq C \Rightarrow C' \subseteq A'$.*
- *$A \subseteq A''$.*
- *$A = A'''$.*

*$C'$ is a set of all attributes shared by the objects from $C$.*

**Definition 4.** *Formal Concept (A, B) is defined as*

- *$A \subseteq G$.*
- *$A' = B$.*
- *$B \subseteq M$.*
- *$B' = A$.*

*A is the concept known as the extent, and B is the concept known as the intent.*

**Definition 5.** *Properties of Closure Operator - (.)''*

- *Monotonicity : $A \subseteq C$ and $A'' \subseteq C''$.*
- *Extensivity: $A \subseteq A''$. A is always a subset of $A''$.*

- *Idempotency: A″″=A″.*

  Example: A closure operator $\partial$ on S is a function assigning a closure $\partial X \subseteq S$ to every $X \subseteq S$ that is
- Monotone: $X \subseteq Y \Rightarrow \partial X \subseteq \partial Y$.
- Extensive: $X \subseteq \partial X$.
- Idempotent: $\partial X = \partial X$.

$(.)″: 2^G \rightarrow 2^G$ and $(.)″: 2^M \rightarrow 2^M$ closure operator. An object subset $A \subseteq G$ closed in (G, M, I) if $A = A″$. An attribute subset $B \subseteq M$ closed in (G, M, I) if $B=B″$.

**Definition 6.** *Properties of sub-concept $\leq$: Let $(A_1, B_1)$ and $(A_2, B_2)$ be two formal concepts of (G, M, I). $(A_1, B_1)$ is a sub-concept of $(A_2, B_2)$ if $A_1 \subseteq A_2$ and $B_1 \subseteq B_2$. $(A_1, B_1) \leq (A_2, B_2)$.*

1. *Reflexivity: (A,B) $\leq$ (A,B).*
2. *Anti-symmetry:If $(A_1, B_1) \leq (A_2, B_2)$ and $(A_2, B_2) \leq (A_1, B_1)$*
   *$\therefore (A_1, B_1) = (A_2, B_2)$.*
3. *Transitivity:If $(A_1, B_1) \leq (A_2, B_2) \leq (A_3, B_3)$.*
   *Then $(A_1, B_1) \leq (A_3, B_3)$.*

A formal concept analysis [41,42] is a method for examining data and knowledge that employs mathematical theory to classify and discover concepts based on lattice theory. It takes a human-centered approach and uses concept lattices to perform exploratory operations. Moreover, it is a framework for data analysis that employs the concept of formal contexts to generate concepts that are then used to represent concept lattices.

A concept, according to its online definition, is "an idea or mental image which corresponds to some distinct entity or class of entities, or its essential features, or determines the application of a term (especially a predicate), and, thus, plays a part in the use of reason or language". A formal concept is derived from the data provided by a formal context.

Since the generated concepts help make concept lattices, this method is quite useful for data interpretation and visualization, as well as the classification of data. A formal concept is defined as a pair of (A, B) where A is a set of objects known as the extent. Extent A consists of all objects that share an attribute in B. B is a set of objects known as the intent. The intent consists of all the attributes shared by the objects in A.

A formal context is a tabular representation of objects and attributes and their heterogeneous relationship. A heterogeneous relation is a binary relation, a subset of the Cartesian product A × B, where A and B are two distinct sets. The following is an example.

A × B gives us a table known as the formal context, where '#' represents 1 in the binary system, i.e., the object has the given attribute. Blank represents that the specific attribute is not present or related to that object. This formal context is used to generate concepts using closure operations and various algorithms explained in the methodology section.

Table 2 is an example of the input known as the formal context. The data are snippets of the original dataset used here. Here, the two distinct sets are:

1. A = {Adani Port, Bajaj Auto, Bajaj Finance, Bajaj Finserv, Bharti Airtel, Infosys}, a distinct set of objects.
2. B = {High-M cap, mid-M cap, low-M cap, very positive ROE, positive ROE, negative ROE}, a distinct set of attributes of the market cap and ROE.

As explained earlier, a formal concept plays a vital role in understanding the data. For the above Table 2, the formal concepts generated are shown in Table 3 below.

**Table 2.** Toy dataset showing the mapping function from eight stocks to six financial ratios.

| | $High_{M_{cap}}$ | $Mid_{M_{cap}}$ | $Low_{M_{cap}}$ | $++_{ROE}$ | $+_{ROE}$ | $-_{ROE}$ |
|---|---|---|---|---|---|---|
| Adani Port | | | # | | # | |
| Bajaj Auto | | | # | | # | |
| Bajaj Finance | # | | | # | | |
| Bajaj Finserv | | | # | | # | |
| Bharti Airtel | # | | | | | # |
| Infosys | # | | | | # | |

**Table 3.** Formal concepts generated for the toy dataset.

| Sr. No. | Objects | Attributes |
|---|---|---|
| 1. | Adani Port, Bajaj Auto, Bajaj Finance, Bajaj Finserv, Bajaj Airtel, Infosys | $\phi$ |
| 2. | Bajaj Airtel | $High\text{-}\kappa_{MC}\ -_{ROE}$ |
| 3. | Adani Port, Bajaj Auto, Bajaj Finserv | $Low\text{-}\kappa_{MC},\ +_{ROE}$ |
| 4. | Bajaj Finserv, Bajaj Airtel, Infosys | $High\text{-}\kappa_{MC}$ |
| 5. | Adani Port, Bajaj Auto, Bajaj Finance, Bajaj Finserv, Infosys | $+_{ROE}$ |
| 6. | Bajaj Finserv, Infosys | $High\text{-}\kappa_{MC},\ \text{Positive}\ \psi_{ROE}$ |
| 7. | $\phi$ | $Low\text{-}\kappa_{MC},\ Mid\text{-}\kappa_{MC},\ High\text{-}\kappa_{MC},$ $++_{ROE},\ +_{ROE},\ -_{ROE}$ |

Table 3 shows the concepts generated in Table 2. Observing the concepts, we can see that seven different concepts are generated.

1. Concept 1 is generated by taking the closure of the object set A, and since the objects have no attributes in common, their closure is an empty set of attributes.
2. Concept 2 is defined for Bharti Airtel, which has a high M cap and negative ROE.
3. Concept 3 is defined for multiple objects, including Adani Ports, Bajaj Auto, and Bajaj Finserv, sharing two attributes: a low-M cap and positive ROE.
4. Similarly, Concepts 4, 5, and 6 are calculated.
5. Concept 7 is defined for an empty object set, which results in an output of the attribute set B.

These concepts are not generated randomly; they follow specific rules and patterns. The process starts by taking the closure of an empty set and then checking for all possible concepts, while simultaneously eliminating redundant and irrelevant concepts.

The generated concepts can be represented using a concept lattice. The concept lattice for the concepts generated above is shown in Figure 3.

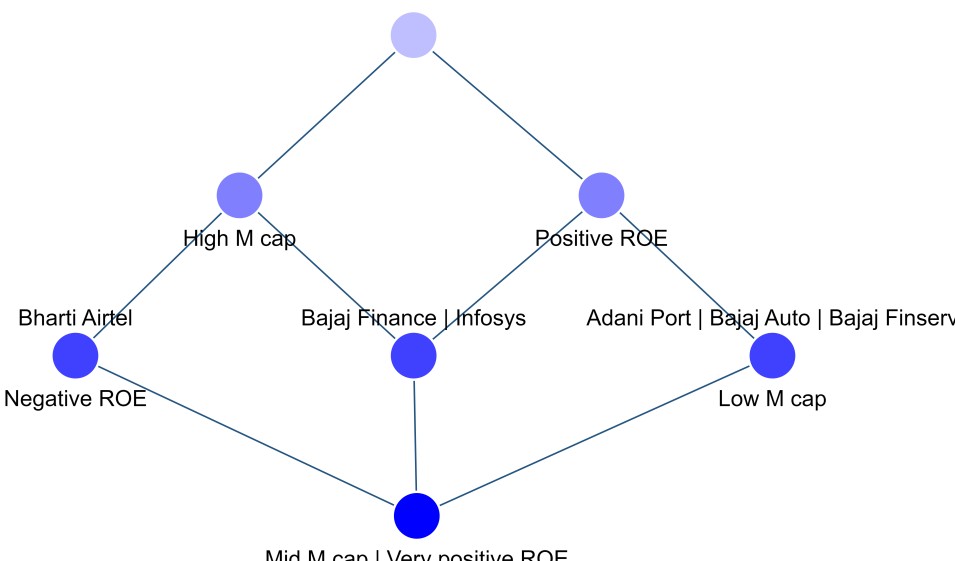

**Figure 3.** Formal concept lattice for the toy dataset.

Reading a lattice is easy; nodes on the same level represent the same degree of stability, while the stability of a node increases as we move up. In this case, the topmost node is empty because, as we saw in Concept 1, there is no attribute that belongs to all objects. Moving to the next level, we see a node for the high-M cap that leads to Bharti Airtel, which also has a negative ROE on the next level. On the same level, we have a positive ROE, belonging to Infosys, Bajaj Finance, Adani Ports, Bajaj Auto, and Bajaj Finserv.

Thus, we can see that the concept lattice corresponds to the earlier identified concepts, but the lattice form is easier to understand and interpret. Moreover, the node below always has the attributes of the node above it, to which it is directly connected.

This is just a short representation of the dataset in real time. The dataset is large, concepts can grow exponentially, and the lattice may become too difficult to comprehend directly.

## 4. Methodology Used

### 4.1. Data Description

Any research or project connected to machine learning relies on data. Data are integral when answering problems.

The dataset used was the NIFTY 50 companies with 15 ratios described in the ratio section. The 15 ratios were
Current market price ($\kappa_{MP}$), Market cap. ($\kappa_{MC}$), P/E ($\psi_{PE}$), D/E ($\psi_{DE}$), ROE ($\psi_{ROE}$), ROA ($\psi_{ROA}$), Asset turnover ratio ($\psi_{ATR}$), Interest coverage ratio ($\psi_{ICR}$), PEG ratio ($\psi_{PEG}$), Dividend yield ($\psi_{DY}$), Quick ratio ($\psi_{QR}$), Current ratio ($\psi_{CR}$), P/S ($\psi_{PS}$), EPS ($\psi_{EPS}$), P/BV ($\psi_{PB}$).

The value of each ratio for each company was collected manually through a Google search and from various websites, using keywords such as "moneycontrol", "finviz", and "screener". The data had to be gathered manually because it is difficult to find the selected ratios or similar datasets directly on the internet. The dataset was collected and stored in an Excel spreadsheet.

### 4.2. Data Prepossessing

The collected dataset needed to be transformed into a binary dataset for the formal concept analysis and scaling was performed manually. For example, a company with a market capitalization of 20,000 crores or more was marked as having a high market capital-

ization, while a market capitalization between 5000 and 20,000 crores was classified as a mid-market capitalization, and a market capitalization below 5000 crores was considered a low market capitalization. As shown in Figure 4, each ratio was divided into value types. For instance, ROE could be very positive, positive, or negative. Thus the dataset was divided into categories for each ratio, giving us a much clearer idea.

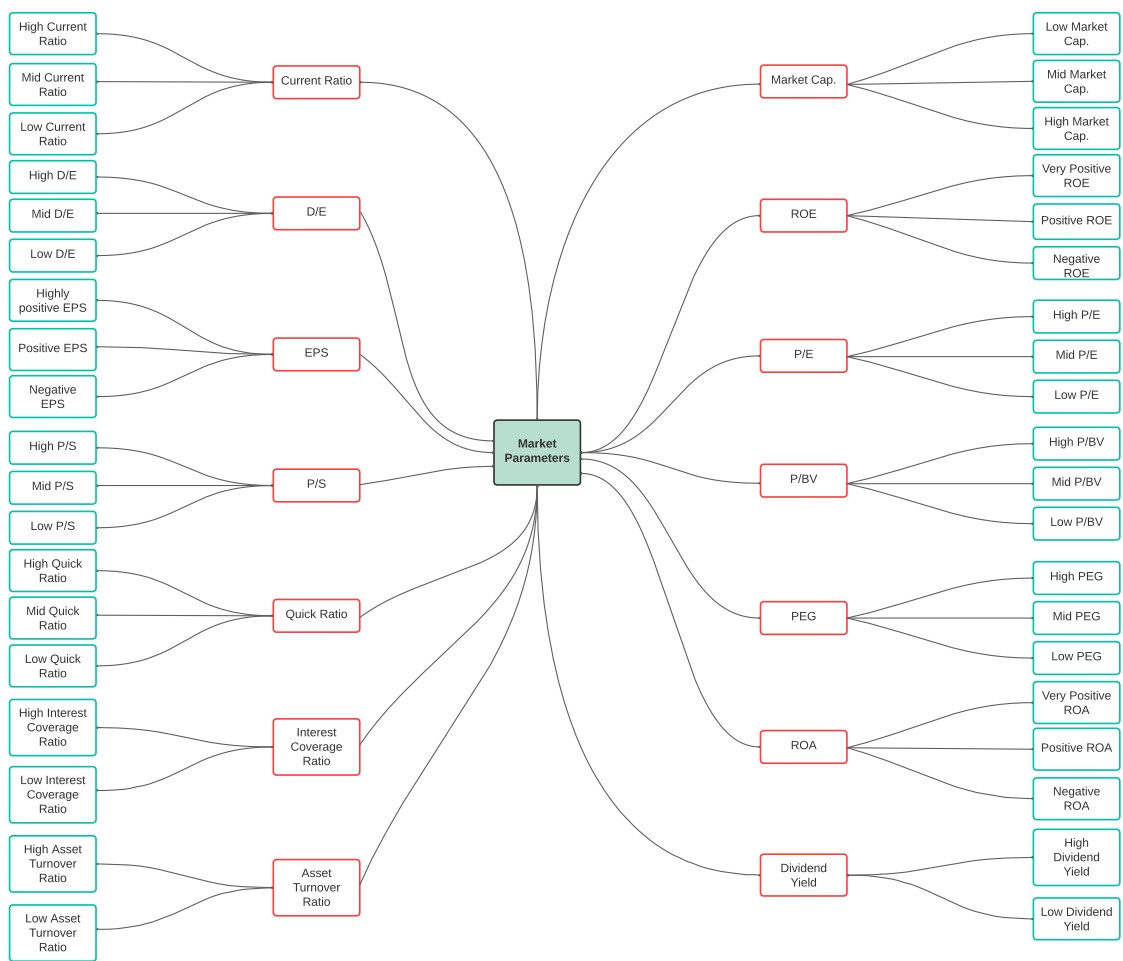

**Figure 4.** Segmentation of parameters of the dataset.

The dataset was transformed from the CSV format to the text file format to make it easier to input into the program. With this format, the FCA algorithm could be applied to the dataset.

### 4.3. Algorithmic Representation

Algorithm 1 represents the steps in evaluating the closure of the set represented by the operator: ''.

Algorithm 2, given below, generates all possible closed sets of the formal context.

The first closure Algorithm 3 given below generates a closed set of an empty set $\phi$.

Next, closure Algorithm 4, given below, generates the lectic next-closed set for the given set.

The stability Algorithm 5 given below calculates the stability index of each set of the lectically closed formal contexts. The set of inputs is (A, B, C), where A and B were calculated above, and C is the concept generated at the end of each next closure algorithm. '|A|' represents the number of elements in A.

---

**Algorithm 1** Closure generation

---

Formulate the closure set of the given input set (A, G, M)
Input: Input structure (A, G, M)
Output: Set of elements forming the closed set $B''$

  1: **procedure** GENERATE—$''$
  2:     B= $\phi$
  3:     **for** each $g \in$ G **do**
  4:         **if** (A $\subseteq$ g$'$)
  5:         Update B $\leftarrow$ B $\cap$ g$'$
  6:     **end for**
  7: **return** B
  8: **end procedure**

---

**Algorithm 2** All closure $(M,'')$ generation

---

Calculate all possible closed sets starting from an empty set A=$\phi$ of a given extent or intent.
Input: A closure operator C $\rightarrow$ C $''$ on the finite linearly ordered set M.
Output: All closed sets in the lectic order.

  1: **procedure** GENERATE—ALL CLOSED SETS
  2:     Update A $\leftarrow$ FirstClosure()
  3:     **while** (A $\neq \phi$) **do**
  4:         Output A
  5:         Update A $\leftarrow$ NextClosure(A,M$''$)
  6:     **end while**
  7: **end procedure**

---

**Algorithm 3** First Closure

---

Calculates the closure of the empty set A =$\phi$.
Input: A closure operator X $\rightarrow$ X$''$ on a finite linearly ordered set
Output: Lectic first-closed set, i.e., closure on the empty set $\phi$.

  1: **procedure** GENERATE—$(\phi)''$
  2:     A= $\phi$
  3:     Update A $\leftarrow$ ClosureGenerator(A,G,M)
  4: **return** A
  5: **end procedure**

---

**Algorithm 4** Closure generation

---

Formulate the closure set of the given input set (A, G, M)
Input: A closure operator X $\rightarrow$ X$''$ on a finite linearly ordered set M and subset A $\subseteq$ M
Output: The lectic next-closed set after A if it exists; otherwise, $\phi$.

  1: **procedure** GENERATE—NEXT CLOSED SET
  2:     **for** each $m \in$ M in reverse order **do**
  3:         **if** ((m $\in$ A)
  4:         Update A $\leftarrow$ A $\cap$ m
  5:         **else**
  6:         Update B $\leftarrow$ ( A $\cup$ m)$''$
  7:         {
  8:         **if** ((B $\cap$ A)>m)
  9:         **return** B
10:         }
11:     **end for**
12: **return** $\phi$
13: **end procedure**

---

---

**Algorithm 5** Stability Calculator

---

Evaluate the stability index of the given input set (A, B, C),
Input: Input structure (A, B, C)
Output: Stability index whose value will be between 0 and 1.

1: **procedure** GENERATE—VARIANCE(A,B)
2:     Var= 0
3:     Update C'← ClosureGenerator(C, G, M)

$$Var = \frac{(C \subseteq A) \cap (C' \subseteq B)}{2^{\|A\|}} \qquad (18)$$

   **return** Var
4: **end procedure**

---

### 4.4. Algorithm Description

The process is divided into two major parts: calculating the concepts and evaluating the stability of each concept.

Calculating the concepts is done in three parts by using two algorithms: first closure and next closure. The closure is a double derivative of a set. Before understanding the algorithm, it is necessary to understand a few concepts of how a derivative and closure are evaluated and what the lectic order is.

Let us understand how the derivative is calculated with an example in Table 4. Given that Table 4 shows a formal context, and as discussed in the above section on the formal concept analysis, it is possible to understand that object P has attributes A, C; object Q has attributes B, C; object R has attributes A, B; and object D has attributes A, B, and C.

1. Suppose the objective is to calculate the first derivative on an empty set of extent E = {}. The " ' " symbol denotes the first derivative.
   (a)     E→E' indicates the intent, i.e., a set of all the attributes common to the set of objects in E.
   (b)     Therefore, an empty set will have all of the attributes common to it.
2. Similarly, calculating the first derivative for a set F = P,S will give us F→F' = {A,C}.

**Table 4.** The toy dataset of the formal context.

|   | *A* | *B* | *C* |
|---|---|---|---|
| P | # |   | # |
| Q |   | # | # |
| R | # | # |   |
| S | # | # | # |

Knowing how the first derivative is calculated, it is possible to calculate the second derivatives on the above data for E' and F'.

1. E' →E'' is the group of all objects common to the set of attributes in E'. E'→E'' = {S}. Since E' = {A, B, C}, which is the attribute set, all of these attributes are common to object S.
2. F'→F'' = {P,S}.

These are the second derivatives. If the second derivative is equal to the original set, then the set is closed and, therefore, considered under the closure operator. From the above calculation, it is possible to see that E →E'→E'' is not a closed set as E''≠ E, whereas F→F'→F'' is a closed set as F'' = F.

The closure operator is valid if the double derivative on a set gives the original set. Closed operations rule out the concepts. This is a Naive algorithm used to calculate all of the possible closures.

There are $2^n$ possible sets of objects and attributes, where n is the total number of objects or attributes. However, as the data grows, the complexity grows exponentially, making it impossible to calculate closure for every set without redundancy. To remove redundancy, the next closure algorithm is used to find the next 'lectically' closed set.

### 4.5. Asymptotic Analysis

Calculating concepts by taking the closure of every permutation and combination of objects is a tedious task. It results in redundancy and, at times, exponential time complexity in the worst-case scenario, as there can be $2^n$ possible concepts. No matter how fast the concept is evaluated, it still takes an exponential amount of time. Therefore, the question is, 'How much time is needed to calculate a closure?'

Let $\alpha$ be the time taken to calculate the closure. Thus, the time complexities for the first closure and next closure algorithm are:

1. First closure : $\alpha$
2. Next closure: $\leq \alpha + O(|M|)$ for one iteration.

The next question that comes to mind is, "What is the value of $\alpha$?" Computing closure(") for a set takes a linear amount of time and is equal to the size of the formal concept, i.e., the size of objects ($|G|$) multiplied by the size of attributes ($|M|$), resulting in $O(|G||M|)$ time complexity for the computing closure.

If $\alpha=O(|G||M|)$, then

1. First closure: $O(|G||M|)$
2. Next closure: $O(|G||M|^2)$

Probing further, it is possible to determine that all closure algorithms work by evaluating the first closure and next closure, and have a polynomial delay as each 'next closure' produces a closed-set polynomial delay. In this case, it is just the complexity of the next closure; therefore, the time complexity for all closures is $O(|G||M|^2)$ when working on objects (concept intent). In some cases, when working on attributes (concept extent), the time complexity for the next-closure and all-closure algorithms is $O(|G|^2|M|)$. However, this may not be the most efficient algorithm for computing concepts. Theoretically, no algorithm that produces concepts can be more efficient, as it takes an exponential amount of time based on context size.

### 4.6. Soundness and Completeness Analysis

To prove the correctness of our algorithm, we state that all the important structures of the context have been taken into account, and nothing unnecessary has been omitted (exhaustiveness property). We assert that our algorithm is correct because it only generates concepts that are structurally similar and does not generate contrived concepts.

In order to do so, we present the notion of 'reflection' on the important object sets inside a given context $\chi(A, G, M)$, while disregarding the rest. Both the number of elements used to generate a reflection and the order in which they appear may provide strikingly different results.

Mathematically, the simple image for a given context $\chi(A_j, G_j, M_j)$ is defined by a set $I \subseteq A_j$, such that

$$\exists g \in G_j \implies I = g' \tag{19}$$

and a collaborative image $I \subseteq A_j$, such that

$$\exists N_i \subseteq \mathbb{N} \wedge \forall g \in G_j \implies I = \bigcap_{i \in N_i} I_i \tag{20}$$

In other words, the object set of the context corresponds to the collaborative image ordering. The ordered set takes $G_j^0$ as the basis and continues to form the $n^{th}$-order with our algorithm. In other words, $I$ is an $n^{th}$-order collaborative image $I \subseteq A_j$ if

$$\forall N_i \subseteq \mathbb{N} \wedge I_{i \in N_i} \tag{21}$$

such that

$$I = \bigcap_{i \in N_i} I_i \wedge \exists i \in N_i \tag{22}$$

In simpler terms, we can establish a one-to-one mapping between the objects of the $n^{th}$-order collaborative image ordering; the algorithm terminates at step $n$. In other words, every concept being created at step $k \leq n$ is an $n^{th}$-order collaborative image ordering.

Each $k^{th}$-order collaborative image $I$ should be a concept extent for a given context $\chi(A_j, G_j, M_j)$.

By the inductive hypothesis, with the base case as k = 0, no objects are taken, which corresponds to $G_j^0$ as the basis of Equations (19) and (20).

Now, if we assume that $I$ has an $n + 1^{th}$-order collaborative image, then $\exists N_i \subseteq \mathbb{N} \wedge \forall g \in G_j$, such that $I = \bigcap_{i \in N_i} I_i$, and each $I_i$ is a $k^{th}$-order collaborative image, with $k \leq n + 1$. We know that the remaining $I_i$ are concept objects generated in previous phases thanks to the inductive premise. This, however, merely implies that $I$ is formed at the latest by step n + 1. To officially show that $I$ could not be formed at a step n + 1, we assume the converse and prove a contradiction.

We assume that $I$ is an $n + 1^{th}$-order collaborative image, which is generated at step $k \leq n$. This means that $I$ also an $n^{th}$-order collaborative image. Hence, $\exists N_i \subseteq \mathbb{N} \wedge \forall g \in G_j$, such that, $I = \bigcap_{i \in N_i} I_i$, and each $I_i$ is an $n^{th}$-order collaborative image (from Equations (19) and (20)). This formally contradicts the fact that $I$ has an $n + 1^{th}$-order collaborative image. This proves the completeness of our algorithm.

In conclusion, all objects in the concepts in the lattice have linkages of either kind above connecting them to the original attribute set. The canonical generator algorithm, or more accurately the objects of the member attributes, play a central role in the crucial chains that 'explain' the origin of $I \subseteq A_j$. As a bonus, even $0^{th}$-order collaborative images may be linked to their own canonical generators, and so on, for all such objects. Here, we focus on identifying the connections between every $I$ and its many generating sets.

## 5. Results and Description

A formal concept analysis, in layman's terms, can be explained as the process of analyzing the data and ruling out concepts that will help considerably reduce risks. The dataset with the algorithms can be found at https://github.com/vaishnavi-88/A-Knowledge-Representation-System-for-Indian-Stock-Market, accessed on 7 April 2023. We performed an initial data statistical analysis of the entire dataset, as shown in Figure 5. The heatmap for the given dataset is displayed in Figure 6. We observe a high degree of correlation between the 'Current Ratio' and 'Quick Ratio'. The diagonal elements of the matrix represent the variance, while the other elements represent the covariance. The variance and covariance are symmetric, as the values to each opposite side of the diagonal are equal, as seen in the heatmap in Figure 6. This may help one understand the nature of the dataset.

| | CMP | Market_Capital | ROE | P/E | P/BV | EPS | D/E | P/S | Current_Ratio | Quick_Ratio | Dividend_Yield | PEG_Ratio | ROA | Interest_Coverage_Ratio | Asset_Turnover_Ratio |
|---|---|---|---|---|---|---|---|---|---|---|---|---|---|---|---|
| count | 50.000000 | 5.000000e+01 | 50.000000 | 50.000000 | 50.000000 | 50.000000 | 50.000000 | 50.000000 | 50.000000 | 50.000000 | 50.000000 | 50.000000 | 50.000000 | 50.000000 | 50.000000 |
| mean | 2750.840000 | 2.837707e+05 | 19.784400 | 33.691600 | 7.384000 | 86.514400 | 1.674600 | 4.119000 | 1.651600 | 1.331000 | 1.524200 | 9.875200 | 8.798000 | 135.864000 | 2.200600 |
| std | 4262.916346 | 3.219859e+05 | 17.650547 | 26.924086 | 13.205488 | 109.113253 | 2.843203 | 3.435381 | 1.261187 | 1.161147 | 1.724021 | 66.376617 | 7.267325 | 655.914251 | 10.517116 |
| min | 132.000000 | 5.539300e+04 | -22.300000 | 0.000000 | 0.640000 | -34.500000 | 0.000000 | 0.000000 | 0.120000 | 0.120000 | 0.000000 | -34.400000 | -3.320000 | 0.000000 | 0.060000 |
| 25% | 606.250000 | 1.000892e+05 | 12.725000 | 14.925000 | 2.267500 | 28.950000 | 0.082500 | 1.582500 | 0.942500 | 0.590000 | 0.450000 | 0.797500 | 3.265000 | 2.242500 | 0.330000 |
| 50% | 1097.000000 | 1.645740e+05 | 15.700000 | 26.050000 | 3.760000 | 55.550000 | 0.565000 | 3.160000 | 1.330000 | 0.915000 | 0.850000 | 1.465000 | 6.900000 | 6.865000 | 0.665000 |
| 75% | 3018.250000 | 3.456252e+05 | 21.275000 | 42.075000 | 6.610000 | 110.250000 | 1.627500 | 5.830000 | 1.832500 | 1.590000 | 1.460000 | 3.145000 | 12.725000 | 42.975000 | 0.975000 |
| max | 20780.000000 | 1.693381e+06 | 113.000000 | 106.000000 | 87.800000 | 646.000000 | 14.800000 | 12.100000 | 6.990000 | 6.390000 | 8.630000 | 467.000000 | 28.800000 | 4605.000000 | 75.000000 |

**Figure 5.** Statistics of the dataset.

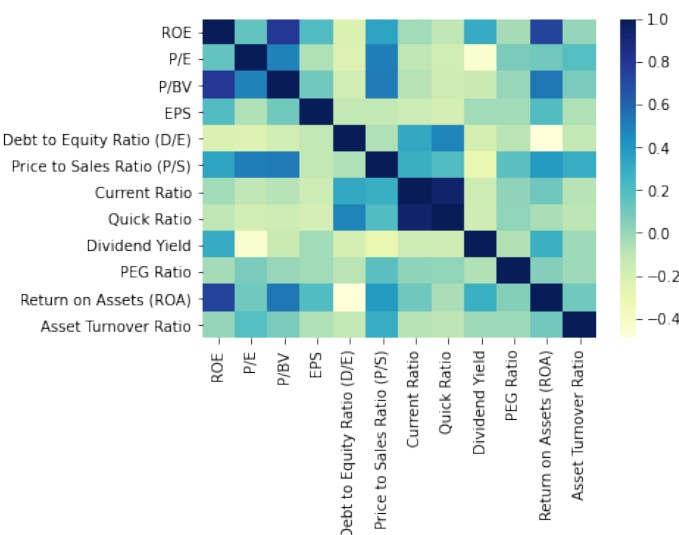

**Figure 6.** Heat map of the original dataset taken.

The dataset was then scaled as shown in Figure 4. The same can be found at https://github.com/vaishnavi-88/A-Knowledge-Representation-System-for-Indian-Stock-Market, accessed on 7 April 2023.

A dummy dataset was used to analyze the expected results, as shown in Table 5.

**Table 5.** Dummy dataset for the analysis of the results.

|  | $High_{M_{cap}}$ | $Mid_{M_{cap}}$ | $Low_{M_{cap}}$ | $++_{ROE}$ | $+_{ROE}$ | $-_{ROE}$ | $High_{PE}$ | $Mid_{PE}$ | $Low_{PE}$ |
|---|---|---|---|---|---|---|---|---|---|
| Adani Ports | # |  |  |  | # |  |  | # |  |
| Apollo Hospitals | # |  |  |  | # |  |  | # |  |
| Asian Paints | # |  |  |  | # |  | # |  |  |
| Axis Bank | # |  |  |  | # |  |  | # |  |
| Bajaj Auto | # |  |  |  | # |  |  | # |  |
| Bajaj Finance | # |  |  |  | # |  | # |  |  |
| Bajaj Finserv | # |  |  |  | # |  |  | # |  |
| Bharti Airtel | # |  |  |  | # |  | # |  |  |
| Bharat Petroleum | # |  |  |  | # |  |  |  | # |

If the above knowledge is compiled and the code is run, one is prompted to enter a value between 0 and 1 to represent the stability of any concept. The stability of a concept is between 0 (least stable) and 1 (most stable).

Enter the threshold value between 0 and 1: 0.5

As one can see, 0.5 was entered as a dummy number for illustration purposes. All concepts with a stability of 0.5 and above are displayed below.

Concept 1: { Adani Ports, Apollo Hospitals, Asian Paints, Axis Bank, Bajaj Auto, Bajaj Finance, Bajaj Finserv, Bharti Airtel, Bharat Petroleum } -> { }
Stability = 1.0000
Concept 2: { Asian Paints, Axis Bank, Bharti Airtel, Bharat Petroleum } -> { }
Stability = 0.5000
Concept 3: { Apollo Hospitals, Asian Paints, Axis Bank, Bajaj Auto Bharti Airtel } -> { }
Stability = 0.5000

Concept 4: { Apollo Hospitals, Asian Paints, Axis Bank, Bajaj Finance, Bharat Petroleum } -> { }
Stability = 0.5000

Concept 5: { Adani Ports, Asian Paints, Axis Bank, Bajaj Finance } -> { }
Stability = 0.5000

Concept 6: { Adani Ports, Apollo Hospitals, Asian Paints, Axis Bank, Bajaj Auto } -> { }
Stability = 0.5000

Concept 7: { Adani Ports, Axis Bank, Bajaj Auto, Bajaj Finance, Bharat Petroleum } -> { }
Stability = 0.5000

Concept 8: { Adani Ports, Asian Paints, Axis Bank, Bajaj Auto, Bajaj Finance, Bharat Petroleum } -> { }
Stability = 0.5000

Concept 9: { Apollo Hospitals, Axis Bank, Bajaj Auto, Bajaj Finance } -> { }
Stability = 0.5000

Concept 10: { Asian Paints, Axis Bank, Bajaj Auto, Bajaj Finance, Bharat Petroleum } -> { }
Stability = 0.5000

Concept 11: { Axis Bank, Bajaj Auto } -> { }
Stability = 0.5000

Concept 12: { Adani Ports, Axis Bank, Bajaj Auto } -> { $High_{PE}$ }
Stability = 0.5000

Concept 13: { Apollo Hospitals, Asian Paints, Bajaj Auto, Bajaj Finance, Bharat Petroleum } -> { $-_{ROE}$ }
Stability = 0.5000

Concept 14: { Adani Ports, Apollo Hospitals, Axis Bank, Bajaj Auto, Bajaj Finserv} -> { $Mid_{M_{cap}}$, $+_{ROE}$, $Mid_{PE}$}
Stability = 1.0000

Concept 15: { } -> { $High_{M_{cap}}$, $Mid_{M_{cap}}$, $Low_{M_{cap}}$, $++_{ROE}$, $+_{ROE}$, $-_{ROE}$, $High_{PE}$ $Mid_{PE}$, $Low_{PE}$}
Stability = 1.0000

One is then asked to enter the number of objects they want to see together, after which, one needs to enter the serial number corresponding to the companies.

| Enter the number of objects: 2 |
| Enter object set to be searched: 2, 5 |

Here, the numbers 2 and 5 represent the serial numbers corresponding to the object, i.e., Apollo Hospitals and Bajaj Auto. Once entered, the list of concepts is displayed along with their attribute sets and what they have in common. Based on this, whether a set has high or low $\psi_{PE}$ ratios, or other such ratios, investors can decide for themselves if they want to invest in the selected set of companies.

| Final Output— |
| Apollo Hospitals Bajaj Auto NOT FOUND |

NOT FOUND means that the two companies do not have a common closure, and they will have to be judged separately.

We can now see that concept analysis takes place by analyzing the object and its corresponding attributes. As a grouping of attributes is formed, it is possible to say that clustering is performed internally. For example, if we look at Concept 14, we can see that the stocks are clustered according to their corresponding attributes. Even if a stock is removed from the index, the concept will still survive, as it has a stability index of 1. Thus, for that concept, we can say that the portfolio manager looking for stocks with the attributes { $Mid_{M_{cap}}$, $+_{ROE}$, $Mid_{PE}$} will obtain the following stocks: { Adani Ports, Apollo Hospitals, Axis Bank, Bajaj Auto, Bajaj Finserv}.

This work can be extended to build a stock market portfolio screener that aims to screen the stocks listed on the stock market exchange and provide end-users with information about certain companies that best fit their investment needs and goals.

Further, the stable concepts generated as a result could be used in deep learning approaches. Knowing the data context is essential for many deep-learning tasks, and this is where knowledge graphs shine. Such associations, such as "Information Technology stocks have High PE ratio", can be recorded in a knowledge representation concept, thus forming a knowledge graph.

Knowledge graphs also help with the prevalent issue of data sparsity in deep learning tasks, which is another advantage. When information is represented as a graph, experts can draw on prior knowledge to fill in the gaps, which is especially helpful when resources are limited.

Knowledge graphs are effective tools for deep learning, especially when it comes to overcoming the restrictions of data sparsity, because they allow for the representation and reasoning of complicated relationships between things.

Understanding and interpreting the logic behind a deep learning model's decisions is called "explainable deep learning"; knowledge graphs can be helpful tools for reaching this goal. Knowledge representation can aid in explaining deep learning. To better incorporate domain knowledge into a deep learning model, it is helpful to represent such knowledge in the form of a graph. The model's precision and readability may both benefit from this.

Understanding the context of information is crucial for many tasks, including natural language processing and recommendation systems, and knowledge graphs can assist a deep learning model in doing so. The model can make better (and more interpretable) decisions if it knows the context. Knowledge graphs are useful for this because they make it possible to see how different items are connected and how the model arrived at its conclusions. This can be especially helpful when the model must make decisions based on intricate interconnections between things. Because of their opaque nature, deep learning models are sometimes referred to as "black boxes," but knowledge graphs can help humans understand the logic underlying the model's inferences and predictions. It is possible to acquire insight into the model's decision-making process by employing the graph to show the connections between things.

When it comes to establishing explainable deep learning, knowledge graphs can be quite helpful thanks to their ability to incorporate domain knowledge, provide contextual understanding, visualize relationships, and interpret black-box models. It is crucial to highlight that different machine learning or deep learning algorithms for categorization can be created on the basis of knowledge graphs provided by domain experts. Our method involves leading the domain expert through the graph construction process by making use of algorithms that extract tuples of knowledge from the data.

## 6. Conclusions

Predicting the stock price of a well-known corporation is difficult because of the wide variety of influences on stock prices. The following are the study's main contributions: the top 50 companies were taken into account according to market capitalization, and 15 fundamental ratios were collected. After preprocessing, we devised a unique feature combination closure algorithm that considered the variety of objects and relationships to build a feature mapping vector for each tuple. The feature combination model was used to generate a knowledge graph in the form of a lattice.

It is of the utmost importance to emphasize the fact that various machine learning or deep learning algorithms for classification have been developed on the basis of knowledge graphs supplied by subject matter experts. Using algorithms that can extract tuples of knowledge from the data, our technique entails guiding a domain expert through the process of building a graph. This is done with the help of the aforementioned algorithms.

Knowledge representation, and graph-based stock market forecasts for well-known companies are intriguing subjects for business. Due to the novelty of the knowledge

graph for feature-set research, it is believed that it will achieve widespread adoption in the academic community. Including a knowledge graph as a feature can help more businesses succeed financially and open up new avenues for growth.

n the future, we plan to use the stable tuples generated from our algorithms to develop an explainable deep neural network using graph embedding for stock price prediction.

**Author Contributions:** Conceptualization, B.P.B., A.R.C., and V.J.; methodology, V.J., B.P.B., and A.R.C.; software, B.P.B., A.R.C.; validation, B.P.B., A.R.C., and V.J.; formal analysis, B.P.B., A.R.C.; investigation, B.P.B., A.R.C.; resources, B.P.B., A.R.C.; data curation, B.P.B., A.R.C.; writing—original draft preparation, V.J.; writing—review and editing, B.P.B., V.J., and A.R.C.; visualization, B.P.B.; supervision, A.R.C. and B.P.B.; project administration, B.P.B., A.R.C., and V.J.; funding acquisition, A.R.C. All authors have read and agreed to the published version of the manuscript.

**Funding:** This research received no external funding.

**Institutional Review Board Statement:** Not applicable.

**Informed Consent Statement:** Not applicable.

**Data Availability Statement:** We have made the data and codes publicly available at https://github.com/vaishnavi-88/A-Knowledge-Representation-System-for-Indian-Stock-Market, accessed on 7 April 2023.

**Acknowledgments:** This work is supported by the "ADI 2022" project funded by IDEX Paris-Saclay, ANR-11-IDEX-0003-02.

**Conflicts of Interest:** The authors declare no conflicts of interest.

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
