# Peer review of "A Knowledge Representation System for the Indian Stock Market"

_computers, doi:10.3390/computers12050090_

Round 1

Reviewer 1 Report

Keywords are found which are insufficiently covered by the authors research. for example: Fundamental Analysis. Other keywords like AI are used in too general sense in sentences where exact/not too general terms should be explored.

1. A collection of terms is included in section 3.1, and this is mentioned as 'and 15 fundamental ratios are collected'. Nothing novel is found in this direction and it is unknown why the keyword 'fundamental analysis' is used in this research. 

2. The Knowledge Representation part  is not novel, this is one of the well-known old and inefficient schemes, and the best close analogs are not included in the overview analysis.  

3. The following part of Conclusions is insufficiently described in the experimental part of the proposed article:

l.678 'This study introduced an innovative approach to forecasting the stock market by embedding knowledge graphs.' 

4. As a whole, the authors didn't compare their own results to Machine Learning and other applications included in the overview part. Neither efficiency nor accuracy or algorithmic complexity problems have been considered. As a result, the presented research is constructed like an idea for an ongoing project where the practical results will appear after months or years of work. 

5. The application of transformers, Generative ANNs, Deep Reinforcement Learning (DRL) and/or other perspective methods and/or technologies stays out of the presented description. This will diminish the potential interest of the readers to the presented research.

6. In Section 5 terms 'clustering' and 'prediction' have been used but only few results of graph operations type 'object X has/has not a path to object Y' are presented. Obviously this Section should be extended and improved, and comparisons [accuracy, complexity, etc.] to best analogs should be included.

Again, here a lot of manual work is detected where DRL and other analogs should present much better results.

Author Response

Response to Reviewer 1 Comments

Point 1: A collection of terms is included in section 3.1, and this is mentioned as 'and 15 fundamental ratios are collected'. Nothing novel is found in this direction and it is unknown why the keyword 'fundamental analysis' is used in this research.

Response 1: At the very onset, the authors would like to thank the reviewer for the valuable time and effort. The ‘fundamental analysis’ term is used on basis of the Figure 2. We are using the ratios and providing a knowledge representation techique in it. It falls under the symbolic AI approach and not statistical AI. Deep learning is a black box model. The authors want to use the stable concepts generated using the algorithms to generate knowledge using lattice. Also this knowledge can farther be used in Deep learning in a context of explainable AI. The same added in the manuscript from line 676-706. 

Point 2: The Knowledge Representation part is not novel, this is one of the well-known old and inefficient schemes, and the best close analogs are not included in the overview analysis. 

Response 2: The authors thank the reviwer for the comment. The Knowledge Representation medhod produces ‘stable’ concepts in the form of a tuple (stocks, attributes) in a heirarchical order of a lattice. This information (concepts) could be labeled as ‘"Information Technology stocks

have High PE ratio’ to from knowledge graph and finally could be used for prediction. As we are not dealing with machine learning (classification), evaluation metrices like accuracy was avoided. The same will be used when a deep leaning architecture will be provided based on the knowledge graph produced as a resultant of this paper which is a future study. In the best close analogs, the domain expert directly provide the knowledge graph whereas our paper is dealing with the technique for knowledge representation to assist the domain expert in order to produce the knowledge graph. Also, the related knowledge graphs donot have the concept of stability. However, based on the comment, we have included some of the latest papers on knowledge graphs in section 2.6 for completeness. It is crucial to highlight that different machine learning or deep learning algorithms for categorization were created on the basis of knowledge graphs provided by domain experts. Our method involves leading the domain expert through the graph construction process by making use of algorithms that extract tuples of knowledge from the data.   

Point 3: The following part of Conclusions is insufficiently described in the experimental part of the proposed article:

l.678 'This study introduced an innovative approach to forecasting the stock market by embedding knowledge graphs.'

Response 3: The authors agree with the reviewer’s comment and made changes in the conclusion part to support the experiment.

Point 4: As a whole, the authors didn't compare their own results to Machine Learning and other applications included in the overview part. Neither efficiency nor accuracy or algorithmic complexity problems have been considered. As a result, the presented research is constructed like an idea for an ongoing project where the practical results will appear after months or years of work.

Response 4: The authors agree with the reviewer’s comment that explainable deep learning is the future study of this article. A proper complexity analysis of our algorithms were shown in the paper in section 4.5. As we are not dealing with machine learning (classification), evaluation metrices like accuracy was avoided. The same will be used when a deep leaning architecture will be provided based on the knowledge graph produced as a resultant of this paper which is a future study.

Point 5: The application of transformers, Generative ANNs, Deep Reinforcement Learning (DRL) and/or other perspective methods and/or technologies stays out of the presented description. This will diminish the potential interest of the readers to the presented research.

Response 5: The authors agree with the reviewer’s comment that explainable deep learning is the future study of this article. Deep learning is a black box model. The authors want to use the stable concepts generated using the algorithms to generate knowledge using lattice. Also this knowledge can farther be used in Deep learning in a context of explainable AI. Our method involves leading the domain expert through the graph construction process by making use of algorithms that extract tuples of knowledge from the data.   

Point 6: In Section 5 terms 'clustering' and 'prediction' have been used but only few results of graph operations type 'object X has/has not a path to object Y' are presented. Obviously this Section should be extended and improved, and comparisons [accuracy, complexity, etc.] to best analogs should be included.

Again, here a lot of manual work is detected where DRL and other analogs should present much better results.

Response 6: The authors thank the reviewer for the comment. The section is improved by adding the outcome of the work in deep learning (explainable) approaches. As we are preparing for an explainable deep learning model (not black box), the results generated could help us attain the same.   

Reviewer 2 Report

In this paper, the authors propose an experimental campaign concerning the Indian Stock Market. The topic considered by the authors is much investigated in the literature. The authors intend to differ their approach from existing ones by using graph theory and constructing a knowledge graph.

The paper is well organized and both the technical and the experimental parts appear well defined.

The main weakness of the paper concerns the innovativeness of the proposed approach. The authors, in fact, compare their approach in the related literature with many stock market forecasting approaches, even partially related to their own. This part of the related literature should be reduced. Instead, the authors should mention and confront graph-based and behavioral-based approaches, even if these study the behavior of subjects in contexts other than the stock market. For example, I suggest they compare their approach with the ones mentioned in the following papers, "Modeling, Evaluating, and Applying the eWOM power or Reddit Posts," "A Space-Time Framework for Sentiment Scope Analysis in Social Media," as well as well as with other approaches that study the behavior of people in social contexts.

Author Response

Response to Reviewer 2 Comments

Point 1: In this paper, the authors propose an experimental campaign concerning the Indian Stock Market. The topic considered by the authors is much investigated in the literature. The authors intend to differ their approach from existing ones by using graph theory and constructing a knowledge graph.

The paper is well organized and both the technical and the experimental parts appear well defined.

The main weakness of the paper concerns the innovativeness of the proposed approach. The authors, in fact, compare their approach in the related literature with many stock market forecasting approaches, even partially related to their own. This part of the related literature should be reduced. Instead, the authors should mention and confront graph-based and behavioral-based approaches, even if these study the behavior of subjects in contexts other than the stock market. For example, I suggest they compare their approach with the ones mentioned in the following papers, "Modeling, Evaluating, and Applying the eWOM power or Reddit Posts," "A Space-Time Framework for Sentiment Scope Analysis in Social Media," as well as well as with other approaches that study the behavior of people in social contexts.

Response 1: At the very onset, the authors would like to thank the reviewer for the valuable time and effort. Based on the comment, we have included some of the latest papers on knowledge graphs in section 2.6 for completeness.

Round 2

Reviewer 1 Report

RE: Author Response 1: At the very onset, the authors would like to thank the reviewer for the valuable time and effort. The ‘fundamental analysis’ term is used on basis of the Figure 2. We are using the ratios and providing a knowledge representation technique in it. It falls under the symbolic AI approach and not statistical AI. Deep learning is a black box model. The authors want to use the stable concepts generated using the algorithms to generate knowledge using lattice. Also this knowledge can further be used in Deep learning in a context of explainable AI. The same added in the manuscript from line 676-706. 

This research is full of introductory texts and explanations. This is a good style aiming at attracting a broader auditory of readers. For pity, the classical and original research data is mixed, and it is difficult to estimate the originality of the research as a whole. 

The new addition significantly improves the explanation part. On the other hand, what new is done in the  ‘fundamental analysis’ direction is insufficiently described. The proposed symbolic AI approach applications should be carefully tested in following directions: computational complexity, soundness, inconsistency, and so on.

Author Response

Response to Reviewer 1 Comments

Point 1: RE: Author Response 1: At the very onset, the authors would like to thank the reviewer for the valuable time and effort. The ‘fundamental analysis’ term is used on basis of the Figure 2. We are using the ratios and providing a knowledge representation technique in it. It falls under the symbolic AI approach and not statistical AI. Deep learning is a black box model. The authors want to use the stable concepts generated using the algorithms to generate knowledge using lattice. Also this knowledge can further be used in Deep learning in a context of explainable AI. The same added in the manuscript from line 676-706.

This research is full of introductory texts and explanations. This is a good style aiming at attracting a broader auditory of readers. For pity, the classical and original research data is mixed, and it is difficult to estimate the originality of the research as a whole.

The new addition significantly improves the explanation part. On the other hand, what new is done in the  ‘fundamental analysis’ direction is insufficiently described. The proposed symbolic AI approach applications should be carefully tested in following directions: computational complexity, soundness, inconsistency, and so on.

Response 1: At the very onset, the authors would like to thank the reviewer for the valuable time and effort. The authors agree with the reviewer hence subsections on asymtopic analysis (computational complexity) and Soundness and Completeness Analysis are given in section 4.5 and 4.6.

Reviewer 2 Report

The authors have striven to satisfy all my requests. Therefore, in my opinion, the paper can be accepted.

Author Response

The authors would like to thank the reviewer for the valuable time and effort.